# Early Salvage Chemo-Immunotherapy with Irinotecan, Temozolomide and Naxitamab Plus GM-CSF (HITS) for Patients with Primary Refractory High-Risk Neuroblastoma Provide the Best Chance for Long-Term Outcomes

**DOI:** 10.3390/cancers15194837

**Published:** 2023-10-03

**Authors:** Juan Pablo Muñoz, Cristina Larrosa, Saray Chamorro, Sara Perez-Jaume, Margarida Simao, Nazaret Sanchez-Sierra, Amalia Varo, Maite Gorostegui, Alicia Castañeda, Moira Garraus, Sandra Lopez-Miralles, Jaume Mora

**Affiliations:** Pediatric Cancer Center Barcelona, Hospital Sant Joan de Déu, 08950 Barcelona, Spain; juanpablo.munoz@sjd.es (J.P.M.); cristina.larrosa@sjd.es (C.L.); schamorro@hsjdbcn.es (S.C.); sara.perez@sjd.es (S.P.-J.); margarida.simaorafael@sjd.es (M.S.); marianazaret.sanchez@sjd.es (N.S.-S.); amalia.varo@sjd.es (A.V.); maite.gorostegui@sjd.es (M.G.); alicia.castanedah@sjd.es (A.C.); moira.garraus@sjd.es (M.G.); sandra.lopezmiralles@sjd.es (S.L.-M.)

**Keywords:** monoclonal antibody, disialoganglioside, anti-GD2 immunotherapy, childhood cancer, high-risk neuroblastoma, naxitamab, chemo-immunotherapy, refractory neuroblastoma

## Abstract

**Simple Summary:**

Patients with high-risk neuroblastoma who are unable to achieve a complete response to induction chemotherapy are known as primary refractory and have poor outcomes. We investigated the combination of chemotherapy (irinotecan (I) and temozolomide (T)) plus anti-GD2 immunotherapy (naxitamab and sargramostim (GM-CSF)), the so-called HITS, against primary resistant high-risk neuroblastoma. Patients were treated when they had measurable chemo-resistant disease at the end of induction treatment but no evidence of progressive disease. Each cycle of HITS comprised Irinotecan 50 mg/m^2^/day intravenously plus Temozolomide 150 mg/m^2^/day orally (days 1–5); naxitamab 2.25 mg/kg/day IV on days 2, 4, 8 and 10, (total 9 mg/kg or 270 mg/m^2^ per cycle), and GM-CSF 250 mg/m^2^/day subcutaneously, for days 6–10 was used. Thirty-four patients received a median of four cycles. Treatment was outpatient and, overall, well tolerated. Patients treated with HITS immediately after induction failure (cohort 1 or early treatment) had a statistically significant improved survival rate compared to patients treated late (cohort 2). In summary, naxitamab-based chemo-immunotherapy is effective against primary chemo-resistant neuroblastoma, significantly improving long-term outcomes when administered early during the course of treatment.

**Abstract:**

Patients with high-risk neuroblastoma (HR-NB) who are unable to achieve a complete response (CR) to induction therapy have worse outcomes. We investigated the combination of humanized anti-GD2 mAb naxitamab (Hu3F8), irinotecan (I), temozolomide (T), and sargramostim (GM-CSF)—HITS—against primary resistant HR-NB. Eligibility criteria included having a measurable chemo-resistant disease at the end of induction (EOI) treatment. Patients were excluded if they had progressive disease (PD) during induction. Prior anti-GD2 mAb and/or I/T therapy was permitted. Each cycle, administered four weeks apart, comprised Irinotecan 50 mg/m^2^/day intravenously (IV) plus Temozolomide 150 mg/m^2^/day orally (days 1–5); naxitamab 2.25 mg/kg/day IV on days 2, 4, 8 and 10, (total 9 mg/kg or 270 mg/m^2^ per cycle), and GM-CSF 250 mg/m^2^/day subcutaneously was used (days 6–10). Toxicity was measured using CTCAE v4.0 and responses through the modified International Neuroblastoma Response Criteria (INRC). Thirty-four patients (median age at treatment initiation, 4.9 years) received 164 (median 4; 1–12) HITS cycles. Toxicities included myelosuppression and diarrhea, which was expected with I/T, and pain and hypertension, expected with naxitamab. Grade ≥3-related toxicities occurred in 29 (85%) of the 34 patients; treatment was outpatient. The best responses were CR = 29% (n = 10); PR = 3% (n = 1); SD = 53% (n = 18); PD = 5% (n = 5). For cohort 1 (early treatment), the best responses were CR = 47% (n = 8) and SD = 53% (n = 9). In cohort 2 (late treatment), the best responses were CR = 12% (n = 2); PR = 6% (n = 1); SD = 53% (n = 9); and PD = 29% (n = 5). Cohort 1 had a 3-year OS of 84.8% and EFS 54.4%, which are statistically significant improvements (EFS *p* = 0.0041 and OS *p* = 0.0037) compared to cohort 2. In conclusion, naxitamab-based chemo-immunotherapy is effective against primary chemo-resistant HR-NB, increasing long-term outcomes when administered early during the course of treatment.

## 1. Introduction

High-risk neuroblastoma (HR-NB) represents approximately half of all NB patients. Even with the current standard of care for HR-NB, including high-dose chemotherapy (HDC) and surgery (induction phase), followed by hematopoietic autologous stem cell transplantation (ASCT) and radiation (consolidation phase), and anti-GD2 immunotherapy and retinoic acid (maintenance phase), the five-year event-free survival (EFS) remains below 60% [1]. As such, and since neuroblastoma is the most common extracranial solid tumor, it accounts for 15% of all pediatric cancer deaths in high-income countries [2].

Conventional therapy for HR-NB includes an initial 5–7 cycles of intensive, multi-agent HDC and surgery, the so-called induction phase, and this response to induction correlates with long-term outcomes. Indeed, inferior EFS was reported by the Children’s Oncology Group (COG) for patients treated in the A3973 clinical trial with post-induction meta-iodobenzylguanidine (MIBG) Curie scores of >2 [3]. Similarly, SIOPEN patients treated in the HR-NBL1 trial with post-induction MIBG skeletal scores of >3 had worse outcomes [4]. The overall COG experience describes how less than a partial response (PR) at the end of induction (EOI) is associated with significantly worse EFS and overall survival (OS) [5]. Consequently, a major goal in HR-NB induction therapy is to achieve the best objective response possible and is ideally a complete response (CR).

Despite intensive induction HDC, 50% of children do not attain CR or very good partial response (VGPR) and are not in remission prior to consolidation therapy [6,7,8,9]. Current results indicate that improving therapy for the induction of remission is urgently needed for this disease. Treatment options for non-responding primary chemo-refractory HR-NB patients are limited, and thus, long-term survival remains poor [3,5]. Several strategies have been tested, including so-called “bridge” therapies, to improve end-of-induction (EOI) responses for these patients with primary refractory disease before entering the consolidation phase of treatment. These include radiolabeled MIBG (131I-MIBG) [10] and combinations of chemotherapeutic agents like topotecan, topotecan plus cyclophosphamide, topotecan plus doxorubicin and vincristine, irinotecan and temozolomide, and irinotecan alone [11,12,13,14,15,16]. Most recently, anti-GD2 monoclonal antibodies (mAbs) have been formally tested in the setting of primary refractory disease, and the results show the efficacy of the combination of dinutuximab with irinotecan and temozolomide (DIT) [17,18] and naxitamab for patients with primary refractory disease in the bone/bone marrow compartment [19]. The early results from using dinutuximab beta with chemotherapy also show objective responses for primary refractory neuroblastoma [20,21].

Early studies from the Memorial Sloan Kettering (MSK) group showed no benefit of adding mAb mu3F8 during induction to improve the high response achieved with HDC alone [22]. However, the most recent study from St. Jude (NB2012), which added anti-GD2 mAb hu14.18K322A to the Children’s Oncology Group (COG) induction chemotherapy backbone resulted in an EOI VGPR rate of 67.7%: a significant improvement on previous COG studies according to authors [23]. However, similar response rates have been seen in prior chemotherapy-only studies like ANBL02P1 [6]. Currently, other cooperative groups are evaluating the benefit of the early use of anti-GD2 mAb therapy during the induction phase of treatment.

Previously, we reported our initial experience using naxitamab in combination with Irinotecan and Temozolomide (I/T) according to the HITS pilot study in relapsed and refractory HR-NB subjects [24]. Like other studies, [17,18,20,21] we reported responses in both refractory and relapse cases. Also, and unlike other studies, we documented responses in subjects who had previously received I/T and/or anti-GD2 immunotherapy. In this study, we evaluated the efficacy, toxicity, and survival among HR-NB patients with a primary refractory disease managed by the HITS regimen. Our results unveil the importance of early salvage chemo-immunotherapy after incomplete EOI responses as it results in increased rates of complete responses and significantly better long-term survival for primary refractory HR-NB patients.

## 2. Materials and Methods

### 2.1. Patient Population

We carried out a retrospective analysis of prospectively collected data of patients who received the regimen HITS (naxitamab plus GM-CSF combined with I/T) from April 2018 to January 2023 for confirmed non-progressive primary refractory HR-NB under compassionate use at HSJD.

Patients were classified as high-risk NB on the basis of standardized criteria, i.e., patients with an INRG stage M diagnosed above 18 months of age or patients at any INRG stage with MYCN amplification [25]. Responses to induction and post-induction treatments were determined according to the 2017 International Neuroblastoma Response Criteria (INRC) [26]. The list of induction regimens includes N7 and mN7, SIOPEN HR-NBL1, GPOH NB2004, Chinese BCC trials, GALOP, COG 3973, and CCG regimens. Patients were classified as “Primary Refractory” if they achieved an incomplete response, partial response (PR), mixed response (MR), or stable disease (SD)) per INRC [26] at EOI. Patients were fully evaluated to discard progressive disease (PD). Disease status was assessed by histology of bone marrow (BM) aspirates from bilateral posterior and anterior iliac crests, a ^123^I- MIBG SPECT scan, and whole-body magnetic resonance imaging (MRI). Fluorodeoxyglucose (FDG)-positron emission tomography (PET) was used for MIBG non-avid cases at diagnosis.

Major organ toxicity was graded according to the Common Terminology Criteria for Adverse Events (CTCAE) version 4.0. Adverse Events (AEs) were prospectively collected and reported to the authorities and Ymabs, according to the established agreements between HSJD and Ymabs Therapeutics. Informed written consent for treatments and tests was obtained according to HSJD institutional rules.

### 2.2. Treatment and Disease Evaluation

Each cycle of chemo-immunotherapy comprised Irinotecan 50 mg/m^2^/day intravenously (IV) plus Temozolomide at 150 mg/m^2^/day orally (days 1–5); naxitamab 2.25 mg/kg/day IV was infused intravenously over 30–60 min, as tolerated, on days 2, 4, 9 and 11, and GM-CSF 250 mcg/m^2^/day subcutaneously, on days 6–10. Treatment cycles were repeated every 4 weeks, and all were administered as outpatients.

Before naxitamab infusion, patients received premedication with an anti-histamine and opioids or ketamine to mitigate pain and infusion-related adverse events (AEs), as described in detail by Mora et al., 2022 [27].

Four BM aspirates and ^123^I-MIBG/FDG-PET scans were performed every 2 cycles in all patients to assess their responses during treatment. Treatment could be continued if PD was not documented at any time. If CR was achieved, then 3–5 more cycles were administered as a consolidation.

A quantitative reverse transcription-polymerase chain reaction was used to assess minimal residual disease (MRD), as described [28] in the BM samples before and after every two cycles of chemo-immunotherapy. During follow-up, disease status was assessed every 3 months for 2 years by the histology of BM aspirates (×4) and MRD plus ^123^I-MIBG/FDG-PET scans. Craniospinal MRI was performed once every year.

### 2.3. Statistical Analysis

Continuous variables were described using the median, minimum, and maximum. Categorical variables were described using absolute frequencies and percentages. EFS was defined as the time from the first HITS treatment to progressive disease (PD), relapse, secondary malignancy, or death and was censored at the last follow-up in the absence of these events. OS was defined as the time from the first HITS treatment to death and was censored at the last follow-up if death did not occur. The Kaplan—Meier method [29] was used to estimate the EFS and OS curves. The prognostic impact of clinical and biological features on EFS and OS was tested using the log-rank test [30]. All *p*-values under 0.05 were considered statistically significant.

## 3. Results

### 3.1. Patient Characteristics and Treatments

The entire cohort consisted of 34 HR-NB patients with a PR or worse (but not PD) at EOI after a full work-up was performed at HSJD. None of the patients had received chemo-immunotherapy of any sort before HITS. The whole cohort profile is described in Table 1.

The patients were categorized into two different cohorts based on the post-induction treatment received. Cohort 1 (n = 17) received no prior anti-GD2 immunotherapy or myeloablative therapy after standard induction and, thus, received early salvage chemo-immunotherapy with HITS (median time from diagnosis to HITS, 8.4 months). Cohort 2 (n = 17) received post-induction therapy and, thus, received late rescue treatment with HITS (median time from diagnosis to HITS, 1.4 years). Post-induction treatments before chemo-immunotherapy included further chemotherapy cycles (cyclophosphamide and topotecan; cyclophosphamide and temozolomide; topotecan-doxorubicin-vincristine—TVD; ifosfamide-carboplatin-etoposide—ICE; topotecan or irinotecan and temozolomide–IT); HDC and ASCT; single-agent anti-GD2 mAbs (dinutuximab family or naxitamab); radiotherapy; and therapeutic MIBG.

The median age of the entire cohort at the time of HITS was 4.9 years. Fifteen (44%) patients were Asian, and 19 (56%) were Caucasian; 12 (35%) were female, and 22 (65%) were male. MYCN amplification was documented in six (17.6%) tumors. Nineteen (56%) patients had received IT prior to HITS, and twelve (35%) received anti-GD2 mAbs (six each, naxitamab and dinutuximab beta) before HITS. Twenty (59%) patients had the disease in the bone and soft tissues; five (15%) soft tissues only; eight (23.5%) in the bone only; and one (3%) with bone marrow disease only. The median size of soft tissue lesions was 30 (range: 0–138) mm, and the median MIBG score was 7 (range: 0–26).

For the early treatment cohort 1 (n = 17), the median age at the time of HITS was 4.9 years. Ten (58%) patients had the disease in the bone and soft tissue; three (17%) had soft tissue only; three (17%) had bone-only disease; and one (6%) had bone marrow disease only. Four (23%) of the 17 had MYCN-amplified tumors. For the late treatment cohort 2 (n = 17), the median age at the time of HITS was 4.8 years. Ten (58%) patients had the disease in the bone and soft tissue (including two with bone marrow disease as well); two (11.7%) were soft tissue only; and five (29%) were bone only. Two (12%) of the seventeen had MYCN-amplified tumors.

A total of 164 cycles of HITS were administered with a median of four (range 1–12), all in the outpatient setting.

### 3.2. Responses to Chemo-Immunotherapy HITS

A summary of HITS responses is shown in Table 2.

Four patients did not complete the first two cycles of HITS, one because of toxicity and three because of early progression, all in cohort 2. For the 30 evaluable patients, responses after two cycles of HITS were CR = 23% (n = 7); PR = 10% (n = 3); and SD = 60% (n = 18). Two (6.6%) patients progressed at the second cycle of evaluation, both from cohort 2. The best responses at any time were CR = 29% (n = 10); PR = 3% (n = 1); SD = 53% (n = 18); and PD = 5% (n = 5). Complete remission was achieved at 2–9 cycles (median = 2; range 1–9). Eventually, three of the ten CR patients relapsed, and seven remained in continued CR; the median follow-up time for alive patients was (n = 20) 27 (range, 3–58) months.

For the early HITS treatment (cohort 1), responses after two cycles were CR = 29% (n = 5); PR = 12% (n = 2); and SD = 59% (n = 10), and best responses at any time were CR = 47% (n = 8); and SD = 53% (n = 9). No disease progression occurred during the treatment of this cohort. Complete responses were achieved at 2–9 cycles (median = 2), and three of the CR patients eventually relapsed. The nine SD patients received 3–6 cycles, with three eventually progressing. At a median follow-up of 27 (range, 3–54) months, 14 (82%) patients were alive while 3 died of the disease.

In the late HITS treatment (cohort 2), one patient was taken off treatment for grade 4 toxicity (see below), and three patients progressed during cycle 1. The best responses were CR = 12% (n = 2); PR = 6% (n = 1); SD = 53% (n = 9); and PD = 29% (n = 5). Complete responses were achieved after two cycles, and none of the two patients relapsed. The eight SD patients received 2–10 cycles, and seven eventually progressed. At a median follow-up of 28 (range, 11–48) months, six (35%) patients were alive, and 11 had died of the disease.

### 3.3. Survival Analysis

Figure 1 shows the Kaplan—Meier curves for the whole population with a 3-year OS of 55.9%, 95% CI = (40.1%, 77.9%) and 4-year OS of 46.6%, 95% CI = (28.6%, 75.9%). Three- and 4-year EFS are 33.8%, 95% CI = (19.8%, 57.5%).

Figure 2 shows the Kaplan—Meier curves for each cohort. A statistically significant improved survival rate both in EFS (*p* = 0.0041) and OS (*p* = 0.0037) is shown for the early vs. late treatment cohorts.

For cohort 1 (early treatment), the 3-year OS is 84.8%, 95% CI = (67.4%, 100.0%), and the 4-year OS is 67.9%, 95% CI = (41.4%, 100.0%). The 3- and 4-year EFS are 54.4%, 95% CI = (32.3%, 91.6%).

For cohort 2 (late treatment), the 3 and 4-year OS are 29.4%, 95% CI = (12.8%, 67.6%). The 3-year EFS is 15.7%, 95% CI = (4.9%, 50.7%), while the 4-year EFS cannot be estimated from the EFS curve.

### 3.4. Safety

The 34 patients completed a median of four naxitamab-based treatment cycles (range 1–12). Severe (mostly grade 3) toxicities occurred in 29 (85%) of the 34 patients, including myelosuppression—either anemia, or low neutrophil or platelet count—(n = 24); hypotension (n = 6); pain (n = 6); hypertension (n = 3); bronchospasm (n = 2); urticaria (n = 1); anorexia (n = 2); diarrhea (n = 3); skin ulceration (n = 1); vomiting (n = 1); laryngeal edema (n = 1); serum sickness (n = 1); a grade 4 increase in alanine aminotransferase and aspartate aminotransferase levels (n = 1); and febrile neutropenia (n = 1). The rates of myelosuppression and diarrhea were expected with I/T, and pain and hyper/hypotension were expected with naxitamab. One patient experienced grade 4 anaphylaxis on day 1 of their first naxitamab infusion and had to stop therapy. No grade 5 toxicity occurred.

## 4. Discussion

Here, we report the detailed use of the chemo-immunotherapy regimen HITS for HR-NB patients with a primary refractory disease at HSJD. Our results show that the HITS regimen is able to achieve CR in 47% of primary refractory HR-NB patients when treated as soon as EOI evaluation determines the failure to achieve CR. In this early treatment cohort, we achieved an encouraging three-year OS of 84.8% and EFS of 54.4%, which is similar to what we recently reported for patients achieving their first CR after standard induction chemotherapy (3-year OS of 81% and EFS of 57%) [31]. These results suggest that incumbent chemo-refractory disease can be overcome when adding naxitamab to rescue chemotherapy early during the course of the disease. These results are significantly better than when HITS was used for patients with protracted primary refractory disease, i.e., those patients who received several rescue regimens, including HDC and ASCT and/or single agent anti-GD2 mAb therapies. Overall, these results highlight how critical it is to promptly add naxitamab during the course of induction to overcome chemo-resistance. The use of HITS for protracted refractory patients is significantly less effective, suggesting that profound and established multi-chemo-resistance prevents the synergism between antibody-dependent cellular cytotoxicity (ADCC) induced by naxitamab and chemotherapeutic agents.

Initial studies showed how the combination of I/T plus anti-GD2 mAbs demonstrated objective responses in patient’s refractory response to chemotherapy and even to single-agent anti-GD2 mAb [17,18,24]. The subsequent question is when to best use this salvage strategy in an already complex multimodality schema of HR-NB management. Our results clearly show that the most favorable outcomes are obtained when HITS is used early during the course of treatment in patients with incomplete responses to standard induction regimens. These results imply that a certain degree of anti-tumor activity from the chemotherapeutic component is critical in order for the combination to work. Although mechanistically, how anti-GD2 mAbs synergize with chemotherapeutic agents is incompletely understood, early preclinical studies suggest that there is an enhanced ADCC through the upregulation of activating stress and inhibitory checkpoint ligands on neuroblastoma cells when using chemotherapeutic agents [32]. Our results suggest that the early use of chemo-immunotherapy before different mechanisms of chemo-resistance develop with the use of multiple lines of salvage chemotherapy, including HDC with ASCT, could be critical to obtaining the best anti-tumor effects with chemo-immunotherapy.

Achieving CR at the end of induction (including chemotherapy and surgery) is the most consistent and reproducible prognostic factor in neuroblastoma [33,34,35,36]. Prior studies demonstrate that chemo-resistance appears early in the course of HR-NB, suggesting that prolonging the time of induction to chemotherapy does not provide a survival benefit. In 1991, Cheung and Heller reported a meta-analysis of 44 trials performed between 1965 and 1989 [37] and concluded that dose intensity had the greatest influence on EFS, while not so much on objective response and OS. Importantly, they showed that the survival benefit plateaued after 21 weeks of treatment, indicating a premature onset of chemo-resistance in the course of HR-NB management. They also predicted that increasing dose intensity would not show a clear improvement in survival, which resulted in serious skepticism of the feasibility of a cure solely using chemotherapy. That prediction sparked the development of treatment alternatives to overcome chemo-resistance, including HDC, ASCT, and anti-GD2 immunotherapy.

Achieving CR is of critical importance for HR-NB patients with *MYCN*-amplified tumors. Actually, it is the only real opportunity to be cured since primary refractory *MYCN*-amplified tumors are virtually incurable in all studies [8,38]. Moreover, different studies, including the MSKCC experience [39] and our own [31], have shown that *MYCN*-amplified tumors in first CR have better survival rates than *MYCN* in non-amplified cases. In this study, we were able to turn two refractory *MYCN*-amplified patients into CR out of the four *MYCN*-amplified tumors managed “early” with HITS, providing these cases with a real chance for long-term survival. Whether much earlier integration of anti-GD2 mAbs in the initial management of *MYCN*-amplified tumors can further increase the CR rates in this subgroup of cases deserves urgent testing.

The relatively low rate of complete responses shown for most induction chemotherapy regimens currently used as the standard in all cooperative groups [6,7,8,9,10,40,41,42,43,44,45,46] confirms the predictions made in the Heller study [37] and highlights how frequently chemo-resistance develops in the early management of HR-NB. Fortunately, chemo-resistant NB seems to be highly responsive to anti-GD2 mAbs and low-dose chemotherapy. Almost half of the primary refractory patients treated with HITS early after chemo-resistance clinically evidenced were able to achieve CR—a necessary condition for long-term survival. Indeed, the survival rates of patients achieving CR after HITS rescue closely resemble those of patients achieving CR with the use of standard induction chemotherapy [31]. Similarly, a recent retrospective analysis by COG showed that treatment with DIT before consolidation with ASCT significantly improved the EFS of primary refractory patients with non-progressive disease [1].

The early use of the anti-GD2 hu14.18K322A mAb, along with induction chemotherapy, generated an improved objective response significantly higher than historical induction regimens [23]. The potential of early anti-GD2 mAb therapy to preempt chemo-resistance and augment CR rates warrants further investigation. Indeed, several ongoing clinical trials are testing the very early use of anti-GD2 mAbs during induction chemotherapy for newly diagnosed HR-NB patients (ClinicalTrials.gov Identifier: NCT05489887; NCT03786783).

Chemo-immunotherapy has become standard management in blood malignancies with extraordinary survival benefits when anti-CD20 mAb rituximab is added to chemotherapy for children and adolescents with high-risk mature B-cell non-Hodgkin’s lymphoma [47]. Incorporating Rituximab into first-line treatment in combination with chemotherapy has paved the way for analogous considerations in pediatric solid tumors. Our results strongly advocate for the early inclusion of anti-GD2 mAb naxitamab in combination with chemotherapy as early as possible during the course of the disease to optimize outcomes.

Fortunately, the toxicity profiles of I/T and anti-GD2 mAbs exhibit no significant overlap. We have shown how naxitamab can be safely combined with I/T in the same way as the dinutuximab family of anti-GD2 mAbs [17,18,23]. The I/T regimen has emerged as the preferred salvage approach as it incurs a lower incidence of hematological toxicity compared to other regimens used as rescue interventions post-induction failure [13,14]. Severe toxicities attributed to naxitamab were observed mostly during infusion and the immediate post-infusion period, as previously reported [26], and its combined administration with I/T did not appear to exacerbate the rate nor the intensity of infusion-related toxicities of naxitamab when observed as a stand-alone treatment.

The integration of additional targeted agents with the induction regimen, such as ALK inhibitors for patients with ALK mutations or mTOR/MAPK inhibitors for those with an activated RAS-MAPK pathway, holds the potential to increase efficacy and broaden salvage alternatives for primary refractory HR-NB patients. Notably, ALK inhibitors have already demonstrated heightened survival benefits among patients harboring ALK mutations with no overlapping toxicities when combined with cyclophosphamide and topotecan [48]. Furthermore, the combination of dasatinib and rapamycin with I/T has exhibited promising results, mainly for patients with *MYCN*-amplified tumors [49]. The potential to improve HITS with the incorporation of these molecularly targeted agents could provide even better chances for primary refractory HR-NB patients to achieve CR and, thus, increase long-term outcomes.

## 5. Conclusions

Naxitamab-based chemo-immunotherapy is effective against primary chemo-resistant HR-NB, increasing long-term outcomes when administered early during the course of treatment. Our results highlight the importance of the early management of chemo-resistance with targeted strategies like anti-GD2 therapy. The synergy between chemotherapeutic agents and anti-GD2 mAbs is mainly seen when chemoresistance is not fully established; therefore, the sooner anti-GD2 mAbs are integrated into chemotherapy regimens, the better chances there are for chemotherapy-enhanced ADCC. The improvement in CR rates at the end of induction with the early use of anti-GD2 mAbs could translate into increased survival rates. This strategy is particularly relevant for *MYCN*-amplified cases.

## Figures and Tables

**Figure 1 cancers-15-04837-f001:**
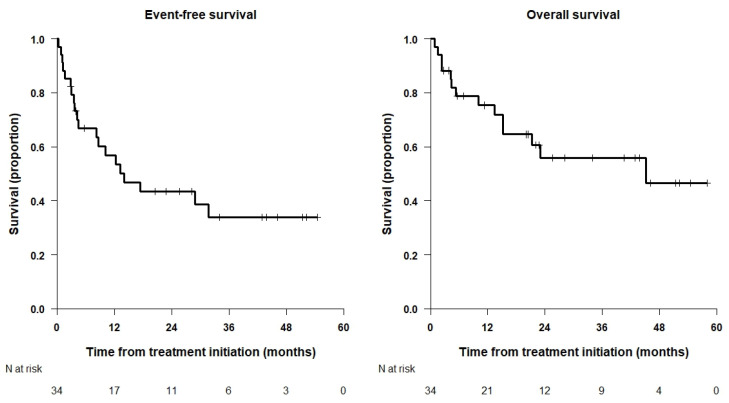
Kaplan—Meier curves for the whole population.

**Figure 2 cancers-15-04837-f002:**
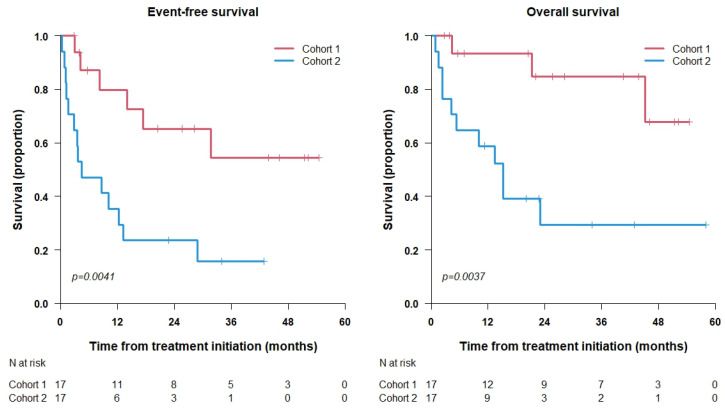
Kaplan—Meier curves for each cohort.

**Table 1 cancers-15-04837-t001:** Summary of the clinical characteristics of all the patients included in the study.

	n = 34	n
Race:		34
Asian	15 (44.1%)	
Caucasian	19 (55.9%)	
Gender:		34
Female	12 (35.3%)	
Male	22 (64.7%)	
MYCN:		34
No amplification	28 (82.4%)	
Amplified	6 (17.6%)	
Prior IT:		34
No	15 (44.1%)	
Yes	19 (55.9%)	
Prior hu3F8:		34
No	28 (82.4%)	
Yes	6 (17.6%)	
Prior to another Ab:		34
No	28 (82.4%)	
Yes	6 (17.6%)	
Age at diagnosis (years)	3.7 [0.7; 27.6]	34
Age at treatment initiation (years)	4.9 [1.8; 33.9]	34
Time from diagnosis to treatment initiation (years)	1.1 [0.2; 6.4]	34
Cohort:		34
1	17 (50.0%)	
2	17 (50.0%)	
Follow-up time for alive patients (months)	26.9 [2.8; 57.9]	20

**Table 2 cancers-15-04837-t002:** HITS responses per cohort.

	Whole Cohort	Cohort #1	Cohort #2
		Early HITS	Late HITS
	n = 34	n = 17	n = 17
**Post 2nd cycle**	n = 30	n = 17	n = 13
CR	7	5	2
PR	3	2	1
SD	18	10	8
PD	2	0	2
**Best response**	n = 34	n = 17	n = 17
CR	10	8	2
PR	1	0	1
SD	18	9	9
PD	5	0	5

## Data Availability

Data sharing is not applicable to this article as no new data were created or analyzed in this study.

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
