# Peer review of "Early Salvage Chemo-Immunotherapy with Irinotecan, Temozolomide and Naxitamab Plus GM-CSF (HITS) for Patients with Primary Refractory High-Risk Neuroblastoma Provide the Best Chance for Long-Term Outcomes"

_cancers, 2023, doi:10.3390/cancers15194837_

Round 1

Reviewer 1 Report

Dear authors, let me stat saying that I found your paper very interested. Highlighting how the chemo-immunotherapy can be a great strategy, also in solid tumour, to treat high-risk neuroblastoma patients that shown resistant to chemotherapy. I have no relevant questions or comments since in the paper your experimental strategy is well described, both the choices of the treatment and the patients selection is well justify as well as the side effects, the results and the overall outcome. 

Author Response

Dear Reviewer,

Thanks for your praise of our work.

Reviewer 2 Report

Sample size is very low to predict any conclusion

Authors mentioned as “Fortunately, the toxicity profiles of I/T and anti-GD2 mAbs exhibit no significant  overlap:. There is no data to support.

Authors mentioned that they fallowed these patients around 24 months, but Kaplan-mayer curves shows up to 60 months, is it a different group?

There are similar studies done previously hence it lacks novelty. Authors need to increase the number of patients so that this study can lead us to a therapeutic regimen.

1. Mora J, Castañeda A, Gorostegui M, Varo A, Perez-Jaume S, Simao M, Muñoz JP, Garraus M, Larrosa C, Salvador N, Lavarino C, Krauel L, Mañe S. Naxitamab Combined with Granulocyte-Macrophage Colony-Stimulating Factor as Consolidation for High-Risk Neuroblastoma Patients in First Complete Remission under Compassionate Use-Updated Outcome Report. Cancers (Basel). 2023 Apr 28;15(9):2535. doi: 10.3390/cancers15092535. PMID: 37174002; PMCID: PMC10177429.

Author Response

  1. Sample size is very low to predict any conclusion

We agree with the reviewer that sample size is limited mainly because neuroblastoma is a rare disease and when we are dealing with specific subgroups (in this case primary refractory disease) numbers go down. Nevertheless, our statistical analysis for comparison is clearly significant therefore, at least we can state that early intervention with HITS provides increased survival compared to late use of chemo-immunotherapy. We do not pretend to predict any conclusion, but have made very concise statements based on what our data can provide.

  1. Authors mentioned as “Fortunately, the toxicity profiles of I/T and anti-GD2 mAbs exhibit no significant overlap:. There is no data to support.

The profile of anti-GD2 antibodies is radically different from any chemotherapeutic agent and this is what the statements means. The profile of naxitamab has been extensively described by us and others.

  1. Authors mentioned that they followed these patients around 24 months, but Kaplan-mayer curves shows up to 60 months, is it a different group?

24 months is the MEDIAN follow-up of patients in this cohort of study. The patient with the longest follow-up is 57.9 months, this is the reason why the KM curves go all the way to 57.9 months.

  1. There are similar studies done previously hence it lacks novelty. Authors need to increase the number of patients so that this study can lead us to a therapeutic regimen. Mora J, Castañeda A, Gorostegui M, Varo A, Perez-Jaume S, Simao M, Muñoz JP, Garraus M, Larrosa C, Salvador N, Lavarino C, Krauel L, Mañe S. Naxitamab Combined with Granulocyte-Macrophage Colony-Stimulating Factor as Consolidation for High-Risk Neuroblastoma Patients in First Complete Remission under Compassionate Use-Updated Outcome Report. Cancers (Basel). 2023 Apr 28;15(9):2535. doi: 10.3390/cancers15092535. PMID: 37174002; PMCID: PMC10177429.

Reviewer cites our own data for the use of only naxitamab (immunotherapy) in patients in first complete response. Here we report the use of chemo-immunotherapy (HITS) for primary refractory patients, a whole different set of patients previously reported with a completely different regimen.

Reviewer 3 Report

The authors properly demonstrated that early intervention with HITS may significantly improve the survival for high-risk NB patients. Overall, this manuscript is well written, and could provide physicians with alternative options to treat resistance high-risk NB. 

Author Response

Dear Reviewer,

Thanks for your praise to our work.

Round 2

Reviewer 2 Report

Authors addressed all the comments